# Mood and Emotions among Inmates after COVID-19 Pandemic

**DOI:** 10.3390/ijerph20064856

**Published:** 2023-03-09

**Authors:** Dorota Chimicz, Agnieszka Lewicka-Zelent, Alicja Lisiecka

**Affiliations:** Faculty of Education and Psychology, Maria Curie-Skłodowska University, 20-612 Lublin, Poland

**Keywords:** mood, emotions, prisoners, COVID-19

## Abstract

Background: During imprisonment, inmates face many difficulties in adapting to life behind bars. The aim of the study was to find out (a) how challenging for inmates were the selected COVID-19 pandemic-related changes and stressors, (b) what moods and emotions are most commonly experienced by prisoners just after the difficult period of the COVID-19 pandemic, and (c) which of the selected factors determine the positive and negative mood of inmates. Methods: The research was conducted in July 2022 in six randomly selected Polish prisons. Prisoners (N = 250) were invited to participate. Comparative and regression analyses were carried out. Two scales measuring moods: the General Mood Scale and the Mood Scale (positive and negative), the Emotions Questionnaire by B. Wojciszke and W. Baryła, and a proprietary questionnaire were used. Results: Sanitary restrictions implemented in prisons resulted in a moderate experience of discomfort among prisoners, mainly in terms of not being able to have direct contact with family and friends, limitation in their personal freedoms to do their jobs, self-development, and deterioration of mental and physical health. A depressed mood predominated among the prisoners, making them feel unhappy, discouraged, tense, and uptight. They reported dominant feelings of alienation, distress, anxiety, and worry at the time of the survey. The mood of inmates was changing from more positive to more negative; on average, it was described as moderate. Based on the regression coefficients, the significant predictors of inmates’ positive mood were perceived happiness (for those who got sick with COVID-19 during their prison sentence) and joy, angst, and contentment (for the healthy ones). In the group of SARS-CoV-2-infected prisoners, unhappiness, age, concern, cheerfulness, and rage were found to be predictors of their negative mood. The feeling of joy appeared to be a significant predictor of negative mood for those inmates who had no personal experience with COVID-19. Conclusions: It is necessary to provide convicts with continuous psychological care and to monitor their mood. Such measures should be the foundation for restorative interventions.

## 1. Introduction

On 11 March 2020, the World Health Organisation (WHO) declared that the COVID-19 outbreak had become a global pandemic [1]. This decision resulted in a range of restrictive measures being implemented across the globe with the aim of reducing the spread of the pathogen. Schools were closed, remote work was introduced, masks were mandated, and social distance was maintained. The restrictions affected everyone, including prison inmates. In almost all European countries, visits were immediately stopped or severely restricted. Educational, sports, and recreational activities were cancelled in most facilities, working hours were suspended, and prison leave was stopped (Table 1). Various types of rehabilitation activities such as addiction therapy, meetings with psychologists, religious services, or the organization of vocational courses and workshops on developing the psychosocial functioning of prisoners were discontinued during the pandemic.

The restrictions were extended over the months of the ongoing pandemic. Their consequences may be felt long after it ends. Assessing the long-term consequences is an important challenge facing the prison system. 

### 1.1. COVID-19 Pandemic in Polish Prisons

As early as the start of the pandemic, World Health Organization (WHO) [2] warned that prisoners were likely to be at greater risk of a coronavirus outbreak than the general population due to the environmental conditions of prisons (e.g., overcrowding, poor ventilation) and the inmates’ health status (e.g., a higher burden of disease related to tobacco abuse, poor hygiene, and weak immune defences due to chronic stress or poor nutrition). The spread of the virus was also thought to be facilitated by regular, necessary routines arising both from legislation and from the need to service the prison and its users. These included the movement of prison staff between the prison and public spaces, the constant introduction of newly incarcerated or detained prisoners, and their transport in multi-person vehicles for judicial, medical, or security purposes [3]. This is compounded by more difficult access to health services than in the community setting [4]. WHO predictions have proven true in prison units around the world, with infection and death rates significantly higher than in the general population [5,6,7].

At the time of the pandemic, international organizations such as WHO, the Committee for the Prevention of Torture (CPT), and The Subcommittee on Prevention of Torture and other Cruel, Inhuman or Degrading Treatment or Punishment (SPT) expressed their concern about the state of the epidemic in prisons by publishing a series of guidelines to be used in prisons. Under these, prison authorities were required to detect COVID-19 early in prisons by providing screening tests; reasonably restrict prisoners’ contact with the outside world (justifying the need for restrictions, as well as providing inmates with information on the virus); equip prisoners with appropriate hygiene supplies; provide those with symptoms of the disease and the medical staff with protective masks; ensure that staff are available to prisoners and provide the former with professional support, healthcare, and training necessary for them to continue their tasks in prison; revise the criteria for the deployment of prisoners and routine grouping situations for prisoners; compensate prisoners for restrictions by increasing opportunities to contact with the loved ones by telephone or Skype; identify prisoners at risk by age or comorbidities; and reduce the number of prisoners [8].

Elementary strategies to prevent and minimize the impact of an outbreak included decarceration (squaring up and releasing prisoners), ensuring physical distance, limiting movement, face covering, cleaning and disinfecting surfaces in communal areas, and training staff on infection control [9]. 

The Prison Service in Poland exercised the law established at the beginning of the pandemic, regulated by the provisions of the Act of 2 March 2020, on specific solutions related to preventing, counteracting, and combating COVID-19 and other infectious diseases, as well as the crisis situations caused by them [10]. Theoretically, the legal Act increased the possibilities of how prisons could respond adequately to the degree of the epidemic threat in penitentiary units and minimize its negative effects. “Possibility” in legal language implies a certain margin of administrative discretion, meaning there is no obligation to act. Taking advantage of the possibilities offered by the COVID-19 Act required additional measures, hiring additional or seconded staff, and using detailed knowledge of prisoners to select the ones who could serve their sentences outside prison.

The rights of convicts during the pandemic were restricted in Poland under the provision of the Executive Penal Code, which allows, in a very general way, to deprive inmates of various rights and privileges. Specific rules and restrictions were decided by prison governors rather than uniform, statutory norms that applied throughout the country. This meant that restrictions looked different in different institutions. The Executive Penal Code empowered governors of penitentiaries and remand centres to suspend or restrict all family, social, work, scientific, sport, and religious activities of the prisoners; order closing cells; and prohibit the possession of certain objects in the cell (Article 247 §1 of the Penal Code). Their decision had to be based on special sanitary or health grounds; they had to specify the duration of the restrictions (up to 7 days) and inform the penitentiary judge. Extending the duration of restrictions or prohibitions required the consent of the latter. The governors could take advantage of this provision by progressively limiting family visits and convict exits from the institution either through outside employment or furloughs. Governors also had the power to place convicts in non-standard (less than 3 square meters) conditions for up to 90 days if an epidemiological or epidemic emergency was declared in the prison’s location or if such an emergency occurred in prison. Such decisions may have been necessary to free cells for quarantine or infirmary purposes. At the end of March 2020, the population in Polish prisons was 91.33%. After analysing the messages posted on the website of each of the 172 prisons, a collective diagnosis was developed by the Helsinki Foundation for Human Rights (HFHR) [11]. Prisons completely restricted family visits, the employment of convicts outside the prison, and religious services or worship. Several prisons restricted collective activities, closed residential cells, stopped walks in semi-open and open prisons, and ceased to grant furloughs. The HFPC’s findings show that 124 units (72%) offered compensation in the form of keeping in touch with relatives through telephone or online communication. In several prisons, mass was broadcasted through a radio station, and three units increased access to hot water [11].

### 1.2. The Nature of Mood and Emotion

According to Larsen [12], mood is one of the primary determinants of human psychological well-being, conditioned by a variety of external and internal factors. Mood is an affective state that is believed to have a pervasive influence on cognitive function and behaviour. It changes slowly as a feedback product of interactions with the social environment [13].

Mood is an affective state typically defined in terms of its slow timescale, integrative properties, and contextual modulation [14,15]. On the one hand, mood is thought to evolve in response to subsequent cognitive appraisals of subjective experiences [12,15,16], and on the other hand, there is evidence that mood alters a wide range of cognitive processes and thus influences the appraisal of future experiences [17,18]. Therefore, at a particular point in time, a person’s mood is both a consequence of their recent actions and a determinant of their future behaviour. The dynamics of this reciprocal interaction have important implications for an individual’s mental health and well-being [19,20].

Bennett, Davidson, and Niv [13] framed a theory of mood according to three general properties of mood: integrative, non-intentional, and contextual. The integrative property means that the valence of mood constitutes a temporal property of the hedonic valence of an experience. However, it does not result from a single event but reflects a changing average of the valences of an individual’s recent experience. There is a strong dependence between the valence of an individual’s mood and the overall valence of their recent experiences: people with recent positive experiences will most likely experience a positive mood, and people with recent negative experiences will experience a negative mood. The non-intentional property states that a good or bad mood (unlike other affective states, such as emotions) is not caused by a specific event. Instead, moods are seen as unfocused and diffuse background states without a specific direction or relation to an object. Philosophy calls this relatedness intentionality, and thus moods are called non-intentional affective states. Mood also has a contextual property. This means the valence of mood depends upon the event’s psychological context”; the same event can impact the mood very differently depending on the context in which it occurs.

According to Wojciszke and Baryła [21], mood is an important psychological factor for several reasons. Firstly, it determines what comes to a person’s mind, influencing the content of their judgments and behaviour. A good mood facilitates the recollection of positive content and enhances evaluations of other people, products, and overall life satisfaction, and strengthens the tendency to help others [22,23,24]. Secondly, it is used as a source of information and a basis for formulating complex judgments using the simplistic heuristic of “how I feel about it”. Instead of considering in detail the various premises of a judgment, we tend to use our own mood as a cue—as long as mood is not discredited as a source of information [25]. The use of the “how I feel” heuristic increases under conditions that force cursory information processing, such as time pressure, making it difficult to challenge mood as a reliable basis for judgment [26]. Third, mood determines whether the information processing is cursory and heuristic or deep, thorough, and deliberate [27,28]. A positive mood informs the mind that the environment is not problematic and does not require any particular action. A negative mood is treated as information that the environment poses problems, and there is a need to take some remedial action. Fourth, mood influences behaviour. A positive mood raises the level of performance in at least some types of tasks, as it enhances motivation via an increase in the subjective attractiveness of the reward gained from the action and an increase in the subjective assessment of the probability of success. Mood also enhances the expectation that one’s own actions will lead to the desired effects [29].

Mood is usually understood as an affective state of moderate intensity, diffuse in character, without a clear reference (subjective cause or specific object at which it is directed), and without a clear physiological arousal component. Discrete emotions (qualitatively distinct), on the other hand, are understood simply as affective states with specific mimetic and pantomimic expressions and clearly defined causes. In contrast with mood, emotions are characterized by relatively strong physiological arousal and relatively short duration. Discrete emotions have the capacity to elicit or reinforce specific, related behaviours (e.g., anger–aggression, tenderness–sacrifice, guilt–satisfaction). The experience of a particular discrete emotion is usually available to a person in the form of a conscious evaluation of one’s own affective state in the form of a judgment such as “I am happy” and “I am embarrassed” [22].

According to Beedie, Terry, and Lane [30], emotion and mood are not the same phenomenon, even though the terms are often used interchangeably. What differs between them is the duration of the experience. Emotions are relatively shorter in duration and moods are longer. Another aspect is that emotions are generally triggered by single events, whereas moods more often result from the accumulation of several events. Then there is timing, with emotions appearing more immediately after the stimuli and mood building up more gradually. The last two elements are control, where emotions are more instinctive and more difficult to control than moods, and intensity, with moods seen as “less intense” compared to emotions.

Despite the differences, researchers note a “transactional relationship” between mood and emotion [31], meaning that in a particular moment, an individual’s mood can impact their emotional response to an event, which in turn can contribute to their mood. Lastly, individual disposition affects how often a person experiences specific moods [32]. 

A subjectively experienced effect, in the form of non-specific moods or specific emotions, is a central strand of human psychological life. Subjective emotions and moods constitute an individual’s quality of life, and for this reason, it is worth measuring them.

### 1.3. The Present Study

The aim of the study was to find out (a) how challenging for inmates were the selected COVID-19 pandemic-related changes and stressors, (b) what moods and emotions are most commonly experienced by prisoners just after the difficult period of the COVID-19 pandemic, and (c) which of the selected factors determine the positive and negative mood of inmates. When we refer to the post-pandemic period, we mean the time when the main sanitary restrictions in prisons and in society were lifted.

We hypothesize that among a variety of restrictions extended over the months of the pandemic, those related to limited activities and freedoms within the already restricted prison environment, as well as deprivation of contact with relatives and friends, were the most burdensome for prisoners. Moreover, lowered mood and negative emotions will prevail in inmates due to two main reasons. Firstly, prolonged additional restrictions within the already restricted prison environment preventing inmates from satisfying many of their needs exacerbated the pains of imprisonment for months. Secondly, concerns about how the virus is being managed both in the prison and in the community where their family members live may cause them to worry about maintaining good health and well-being. The perceived anxiety about one’s own health and that of loved ones should also not be forgotten. The pandemic was associated not only with getting sick but also with the altered functioning of the entire healthcare system and, consequently, with limited access to healthcare. It is, therefore, assumed that the mood of inmates is determined not only by their emotions connected with imprisonment but also by their personal experience of the illness. Finally, negative emotions will condition the negative mood of the inmates, while positive emotions will condition their positive mood.

## 2. Materials and Methods

### 2.1. Participants

The study included people serving prison sentences (*n* = 250). Of the respondents, all identified as male. Ages range between 19 to 77 years. Almost two-thirds of inmates (64% each) were between 25 and 45 years old. Prisoners up to the age of 24 accounted for 11% of the respondents, and seniors aged 60 and over accounted for 6%. One-third of the respondents completed their education at the vocational school level. The same number graduated from elementary school. Five percent of them were college graduates. The remaining men had a high school diploma. Fewer than 3/4 of them (72%) were under quarantine during the pandemic, and 26% of inmates reported their relatives to be isolated after being diagnosed with COVID-19. More than half of the respondents (*n* = 141, 56%) were not infected with SARS-CoV-2. It means that 109 respondents (44%) became sick with COVID-19.

### 2.2. Procedure

The study was conducted in July 2022 in six randomly selected Polish prisons. This was made possible by obtaining permission for the study from the director of the District Inspectorate of Prison Services. The survey was conducted individually with each inmate in a separate room of a Prison Service officer by penitentiary educators or psychologists. The inmates agreed to participate in the study. A detailed informed consent statement was provided to all participants with the right to withdraw, anonymisation, including pseudonyms, and use of data.

### 2.3. Materials

The study used a quantitative strategy. Two standardized survey instruments and a proprietary questionnaire were used. Two scales measuring mood were used: the General Mood Scale (consisting of 10 statements rated on a 5-point scale from 1 to 5) and the Mood Scale (positive and negative) by B. Wojciszke and W. Baryła [21]. The scales are characterized by satisfactory psychometric properties. The authors indicate that Cronbach’s alpha for the General Mood Scale is 0.96 (0.77 for the Negative Mood Scale and 0.78 for the Positive Mood Scale). The discriminatory power of individual items on the General Mood Scale ranges from 0.76 (item 1) to 0.87 (item 6). The stability of the scores after four weeks was 0.46 (for the Negative Mood Scale—0.44 and for the Positive Mood Scale—0.48). The accuracy of the scales was determined by correlation with the Mood Lowering and Mood Elevating Scales (*p* < 0.05). They are used to determine the mood experienced by an individual at the time of the survey and their positive and negative mood persisting over the past week. The third tool used in the study was the Emotions Questionnaire by B. Wojciszke and W. Baryła [21]. It consists of 24 adjectives describing discrete emotions, four for each of the six main emotions: joy, fear, anger, sadness, and guilt. They were taken from E. Diener et al. Respondents rate each emotion on a 7-point scale, where 1 means the person has never experienced it and 7 means they have constantly experienced it in the past week. The Cronbach’s alpha value for the Emotions Questionnaire is as follows: Joy—0.88, Love—0.86, Fear—0.88, Anger—0.75, Guilt—0.75, and Sadness—0.91. The stability of the results after four weeks was: 0.44; 0.61; 0.54; 0.52; 0.51, and 0.55 for each emotion, respectively. The discriminatory power of the individual questionnaire items ranges from 0.75 (guilt) to 0.91 (breakdown and unhappiness). The accuracy for the main emotion was determined by correlation with the Mood Depression and Mood Elevation Scales (*p* < 0.05).

A proprietary questionnaire containing demographic data and questions concerning the severity of selected aspects of COVID-19 pandemic related to nine possible changes and stressors in prison life was also used. Respondents answered questions using the 5-point scale (where 1 meant—“Not hard/worrying” and 5 meant—“Causing severe distress”).

## 3. Results

### 3.1. COVID-19 Pandemic-Related Changes and Stressors

The above literature review shows that prisons are proved to be a ground for infectious diseases due to specific conditions, such as overcrowding, inadequate infrastructure and health care, confined unhygienic spaces with inadequate ventilation, and poor physical and mental health of the prison population. These conditions after the outbreak of the COVID-19 pandemic placed prisoners at a heightened risk of contracting it and dying. Correctional systems worldwide, as an integral part of the public health response to the pandemic, introduced a range of strategies and restrictions to reduce the risks of COVID-19 outbreaks. Therefore, inmates were asked to rate the selected COVID-19 pandemic-related changes and stressors as an inconvenience, difficulty, or causing suffering. They rated the different examples of pandemic consequences on a scale from 1 to 5 points, where 1 mean—I suffered no negative consequences and 5 mean—I felt great discomfort in this respect. Table 2 contains the mean, standard deviation, skewness, and kurtosis.

The values of the average in each statement range from 2 to 3 points. They show that the indicated effects of the pandemic had a minor or moderate impact on prisoners’ everyday life behind bars. Respondents acknowledged that the time of sanitary restrictions resulted in a moderate experience of discomfort/difficulties mainly in terms of not being able to have direct contact with their family and friends, limitation in their personal freedoms to do their jobs, self-develop, e.g., developing their interests, but also regarding the perceived deterioration of their mental health. During the pandemic, deprivations of contact with the Prison Service and fellow co-inmates, the lack of opportunity to participate in volunteering and the impact on their physical health appear to be a burden to a minor extent. 

In order to test whether there were statistically significant differences in the assessment of the negative effects of the COVID-19 pandemic between inmates who got sick and those who did not, the Student *t*-test for independent samples was used. The values of this test and levels of statistical significance are presented in Table 3.

The *t*-test values and the level of statistical significance show that there were statistically significant differences between the means obtained in the different inmates’ groups for four of the nine assessed negative consequences of the pandemic. Inmates who had personally experienced the disease (either themselves or their family members) felt significantly stronger consequences of the pandemic in their interactions with prison officers (Inmates1—M = 2.27, SD = 1.50; co-inmates2—M = 1.70, SD = 1.24) (*p* < 0.001) (d Cohena = 0.3) and co-inmates (Inmates1—M = 2.29, SD = 1.49; Inmates2—M = 1.87, SD = 1.35) (*p* < 0.05) (d Cohena = 0.3). They also acknowledged that they suffered more negative consequences in terms of physical health (Inmates1—M = 2.72, SD = 1.56; Inmates2—M = 2.11, SD = 1.49) (d Cohena 0.4), but also in mental health (Inmates1—M = 2.86, SD = 1.52; Inmates2—M = 2.34, SD = 1.52) (*p* < 0.01) (d Cohena = 0.4). 

### 3.2. Respondents’ Mood 

The first step in analysing the obtained data on the mood of the respondents after the COVID-19 pandemic was to calculate descriptive statistics. Table 4 contains the mean, standard deviation, skewness, and kurtosis. Filling in the General Mood Scale, respondents used a 5-point scale, in which 1 meant “disagree”, 2 “rather disagree”, 3 “a little bit yes, a little bit no”, 4 “rather agree”, 5 “agree”, rating 10 statements describing their mood at the time of the survey. 

Each statement from the General Mood Scale was rated at an average of 3 points. This means that the convicts had doubts when reflecting on their mood. They were not able to clearly answer what kind of mood they were in and how they felt. The result can be interpreted as moderate mood. The values of skewness and kurtosis show that the distribution of the results is normal, as none of them exceeds the value of 1. The result indicates that the respondents felt a moderate mood, neither too positive nor too negative. 

To determine whether there are statistically significant differences in mood scores between convicts who got sick with COVID-19 and those who did not have this infection, a Student’s *t*-test for independent samples was conducted. The values of this test and levels of statistical significance are shown in Table 5.

There was a statistically significant difference in the negative mood scale (*p* < 0.05) between the mean scores obtained by inmates who were (M = 2.83; SD = 3.10) and were not infected with SARS-CoV-2 (M = 2.10; SD = 2.65). This means that the negative mood of inmates who were ill was rated significantly lower by them compared to fellow inmates who were not infected with SARS-CoV-2. The effect size of Cohen’s d = 0.3 is moderate.

### 3.3. Emotions Experienced by the Respondents 

Respondents used the Emotions Questionnaire to rate the frequency of experiencing various emotions and feelings over the past week on a 7-point scale, where: 1 means “never”, 2 “very rarely”, 3 “rarely”, 4 “sometimes”, 5 “often”, 6 “very often”, and 7 “always”. The values of mean, standard deviation, skewness, and kurtosis are given in Table 6.

The obtained data show that convicted persons sometimes felt joy, love, cheerfulness, and contentment, attachment, but also sadness. Very rarely did they feel humiliated. The other emotions appeared infrequently. 

The authors of the Emotion Questionnaire distinguished six main emotions: HAPPINESS, LOVE, FEAR, ANGER, GUILT, and SADNESS, which the subjects rated on a 7-point scale. Each of these emotions included another four specific emotions, meaning that respondents could score from 4 points to 28 points on each emotion. Table 7 shows the obtained results: the mean, standard deviation, skewness, and kurtosis for six main emotions.

Out of the six main emotions listed, inmates most often experienced HAPPINESS (M = 14.15) and LOVE (M = 13.50) for the last days up to the time of the survey, and the least often SADNESS (M = 11.50). The skewness and kurtosis in terms of HAPPINESS and LOVE indicate a negatively skewed distribution (too many results are high and too few are low) and a leptokurtic distribution (results are more concentrated around the mean). 

In order to determine the differences between the mean values obtained in the compared groups for the main emotions, the Student’s *t*-test for independent samples was applied. Table 8 shows the values of the Student’s *t*-test and statistical significance for the main emotions.

The value of the Student’s *t*-test does not indicate statistically significant differences between the mean scores for the six main emotions felt by the detainees in the compared groups. This means that inmates who experienced illness due to COVID-19 rated the emotions experienced similarly to inmates without such experiences.

### 3.4. Determinants of Positive and Negative Mood in Inmates

To determine the mood predictors of inmates, we conducted a linear regression. Sixteen negative and eight positive emotions were included in the model. Table 9 shows the obtained results. 

The model developed for inmates who were infected, in which the criterion variable is negative mood, includes a combination of 25 variables. The F-test value indicates that the model is significant (F (27,109) = 5.222; *p* < 0.001). The variables that were included in the model explain 64% of the variance, as shown by the R^2^ value. Based on the regression coefficients, joy was found to be a significant predictor of inmates’ negative moods. This means that the less they felt joy, the more negative mood they experienced. All variables entered the model. The remaining variables entered the model but did not reach statistical significance.

However, the model developed for inmates who have not personally experienced COVID-19 sickness, in which negative mood is the criterion variable, includes a combination of 25 variables. All variables entered the model. The F-test value indicates that the model is significant (F (4,239) = 6.887; *p* < 0.001). The variables that were included in the model explain 54% of the variance, as shown by the R2 value. Unhappiness, age, concern, cheerfulness, and rage were found to be significant predictors of inmates’ negative moods. This means that the weaker they experienced cheerfulness and rage, the more negative their mood was. In addition, the older they were, the more strongly they felt unhappiness and concern, and the worse their mood was as well. It was also found that the stronger the sadness the respondents felt, the greater the tendency for them to experience negative moods (*p* = 0.065). The remaining variables entered the model but did not reach statistical significance.

Moreover, a linear regression was conducted for the model in which positive mood was the dependent variable. The same variables were included: negative and positive emotions, as well as age. Table 10 shows the obtained results.

The resulting model for inmates who did not have COVID-19, in which a positive mood is the criterion variable, contains 25 variables. The F-test value indicates that the model is significant (F (25,141) = 4.796; *p* < 0.001). The variables that were included in the model explain 64% of the variance, as indicated by the R^2^ value. For positive moods in inmates who were not infected with COVID-19, joy, angst, and contentment appeared to be significant predictors. The more strongly inmates experienced joy and contentment and the less angst, the more positive their mood. It was also found that the weaker the sadness they experienced, the stronger the tendency for them to experience positive mood (*p* = 0.058). The remaining variables entered the model but did not reach statistical significance.

A statistically significant model was obtained for inmates who got sick with COVID-19 (F (2,109) = 8.091; *p* < 0.001). The variables included in the model together explain 62% of the variance. Happiness appeared to be the only significant predictor. Inmates experienced a more positive mood along with experiencing happiness. In addition, a tendency for improved mood was found to occur with a stronger experience of a slump (*p* = 0.057).

## 4. Discussion

During imprisonment, inmates face many difficulties in adapting to prison life, well depicted in either classical works or recent research [33,34,35,36,37]. 

Leśniak and Szaszkiewicz [38] point out that the main inconveniences of incarceration include detention itself (including the feeling of moral rejection by society and placing the prisoner in an inferior social category), deprivation of material and service facilities, sexual contacts, and autonomy (i.e., placing the prisoner in the role of an object under other people’s influence) or, finally, deprivation of the sense of security. From a psychological point of view, these discomforts are aggravated by detachment from the lifestyle (contact with relatives, interruption of activities (employment, leisure, hobbies), withdrawal from social activities, lack of opportunity to make decisions about oneself, the need to stay in a closed space shared with other prisoners, and severe violation of privacy boundaries. This constant state of deprivation of basic needs causes frustration and leads to many intrapsychic conflicts, an excess of negative experiences, and lowered mood. The results of the presented research confirm these conditions. A depressed mood was prevalent in prisoners, making them unhappy, discouraged, tense, and wired. They reported dominant feelings of alienation, distress, anxiety, and grief at the time of the survey. 

The results indicate that the respondents felt a moderate mood, neither too positive nor too negative. Nonetheless, they rated their mood worse than other groups examined by Wojciszke and Baryła [21], i.e., the homeless, civil servants, or students. The mean values in these groups indicated either a moderately positive mood (students and homeless people) or a highly positive mood (civil servants). 

The study also shows that inmates had difficulties assessing their moods and emotions. They were not able to give a clear answer as to what mood they were in. It might indicate that they may have elevated levels of alexithymia [39]. However, this requires further research. Alexithymia is the inability to identify or express emotions. It is expressed through difficulties in identifying feelings, difficulties in describing feelings, and a focus on externally oriented thinking patterns [40]. Related constructs include emotional intelligence, which has been described as a way of quantifying emotion regulation [41] and refers to an individual’s ability to understand, process, and manage emotions, as well as the ability to integrate their effects [42].

The results of the research also showed that the prisoners experienced all the negative emotions distinguished in the Emotion Questionnaire [21], i.e., anxiety, fear, anger, humiliation, sadness, crisis, grief, concern, rage, shame, guilt, depression, and unhappiness (*p* < 0.001). As previous studies show, prison is an emotionally painful place for inmates [43,44], where feelings of anxiety, fear, sadness, hopelessness, frustration, grief, anger, resentment, and depression are experienced daily, whereas joy, hope, satisfaction, and happiness occur much less frequently.

The literature review shows that prisoners are a psychologically vulnerable group with high levels of existing mental health difficulties [45,46,47]. Often problems of this nature are present even before incarceration but are significantly exacerbated during the course of their sentence. The over-representation of psychopathology in the prison context is a reflection of both the vulnerability that these individuals typically exhibit and the lack of appropriate conditions and forms of support offered in prisons. The prison environment can impair inmates’ mental health, well-being, and, consequently, social functioning. Upon entering prison, convicts are often confronted with a new reality, facing a number of challenges. The process of their peculiar enculturation into the prison culture, which usually involves learning and adopting maladaptive “norms” of “the prison code”, might have a negative impact on the adaptation and rehabilitation of the prisoners [48,49].

Although the genesis of prisoners’ mental health difficulties is not a new issue, it is undoubtedly the reality of recent years enforced by the COVID-19 pandemic that has exposed further areas of perceived difficulty. Efforts made to prevent the spread of the virus have resulted in a series of restrictive measures aimed at limiting contact between prisoners and between prisoners and prison staff. The experience of such double isolation (“locked in and locked down”) brought with it negative effects also in the realm of perceived emotions. Sadness and depressive states resulting from the lack of contact with family and other close people were common experiences. Many prisoners indicated feelings of loneliness and frustration that made it difficult to endure isolation during this difficult time [50]. The introduced sanitary restrictions, which shattered the daily schedule associated with work, education, and contact with other people, resulted in an increased sense of helplessness and increased levels of perceived anxiety. Even in countries where prisoners were able to benefit from non-contact visits or communicate with relatives using technology, the lack of physical contact with partners and children had a negative impact on their mental state. Because of the lack of visits and regular communication, many prisoners were worried about their families and how the ongoing pandemic would affect their loved ones. The psychological state of the prisoners was also exacerbated by the pervasive anxiety about the risk of infection in prison. This resulted in increased feelings of anxiety, sleep difficulties, medication or drug abuse, incidents of self-harm, and suicide attempts [51]. An analysis of the prisoners’ emotional and psychological situation during the COVID-19 pandemic thus indicates persistent difficulties in this area even after the restrictions were lifted. 

The wider literature considering emotions recognizes that much more attention is paid to negative states than to positive ones [52]. Despite this, there is evidence to suggest that expressing positive emotions can reverse the detrimental physiological effects of experiencing negative emotions [53,54]. 

The results of the present study have several practical implications: they emphasize the need to develop and implement preventive interventions in which prisoners are taught to use cognitive, adaptive emotion regulation strategies that would prove beneficial to their mental health. These interventions would act as preventive health measures, helping prisoners to develop a diverse set of healthy, constructive ways of coping with various negative effects of adverse situations associated with the prison environment and global challenges, as exemplified by the COVID-19 pandemic. Strategies to enhance emotional resilience should also focus on incorporating psychoeducation on the role of various cognitive emotion regulation strategies in coping with life challenges, both behind prison walls and after leaving prison.

Apart from its strengths, the present study has some limitations. A longitudinal study would capture possible changes in the perceived emotions and mood among inmates, taking into account, among other factors, the climate in a particular prison and the dynamics of the pandemic restrictions. Unfortunately, such a diagnosis was not possible due to the restriction in penitentiary units during the pandemic. Additionally, it would be valuable to capture cultural differences in international studies.

## 5. Conclusions

Experiencing incarceration is undoubtedly not conducive to maintaining a good mood and feeling positive emotions. It is, therefore, necessary to provide prisoners with constant psychological support and to monitor their moods. As Larsen [12] argues, a person’s mood is related to their ability to cope with environmental demands, and as demonstrated above, these are extremely stressful for prisoners. Any neglect of the emotional sphere in persons deprived of liberty may cause irreparable damage disturbing their sense of security, negatively affecting how they build relationships with other people, and inhibiting the process of positive changes in personality and behaviour. Hence, it is important to train inmates in their ability to recognize emotions and cope with unpleasant ones. It could even be argued that this should form the foundation for restorative interventions [55]. We often think of imprisonment as something people go through, and it finally ends when they are released. The reality, however, is that time spent in prisons can result in a range of side effects accompanying prisoners even many years after they stop serving their criminal sentence.

## Figures and Tables

**Table 1 ijerph-20-04856-t001:** Restrictive pandemic measures in prisons around Europe (symbol X indicates the restriction was introduced).

Country	Visits Limited or Stopped	Activities Limited or Stopped	Prison Leave Stopped	Movement in Prisons Restricted
Poland	X	X	X	
France	X	X		X
Germany	X	X	X	X
Italy	X	X		
Spain	X	X	X	X
Czech Republic	X	X	X	
Romania	X	X	X	
Hungary	X	X	X	X
Portugal	X	X	X	
Greece	X		X	
Belgium	X		X	X
Slovak Republic	X		X	
Netherlands	X		X	
Austria	X	X	X	
Bulgaria	X	X	X	
Switzerland	X			
Lithuania	X	X	X	
Albania	X			
Denmark	X	X	X	
Ireland	X			X
Croatia	X	X	X	
Latvia	X	X	X	
Norway	X	X	X	
Finland	X	X		
Estonia	X	X		X
Slovenia	X			
Cyprus	X	X		
Malta	X			
Luxemburg	X	X	X	X

Source: github.com/dw-data/covid-prisons (accessed on 5 October 2022).

**Table 2 ijerph-20-04856-t002:** Descriptive statistics for COVID-19 pandemic-related changes and stressors in inmates’ opinion (N = 250).

Pandemic-Related Changes and Stressors	M	SD	Skewness	Kurtosis
Deprivation of contact with relatives and friends	3.35	1.63	−0.48	−1.20
Deprivation of access to work	2.61	1.65	0.17	−1.26
Deprivation of contact with other inmates	2.06	1.43	0.64	−0.72
Deprivation of contact with prison officers	1.94	1.39	0.72	−0.45
The limited access to self-development services	2.56	1.60	0.08	−1.25
Deprivation of liberty	2.94	1.75	−0.17	−1.45
The limited access to voluntary service	2.01	1.57	0.53	−0.84
Physical health deterioration	2.38	1.55	0.37	−1.15
Mental health deterioration	2.56	1.53	0.22	−1.23

Explanations: M—mean, SD—standard deviation.

**Table 3 ijerph-20-04856-t003:** Differences between mean values for pandemic-related changes and stressors obtained in the group of inmates who were (N = 109) and were not infected with SARS-CoV-2 (N = 141).

Pandemic-Related Changes and Stressors	*t*-Test	*p*
Deprivation of contact with relatives and friends	−1.16	0.266
Deprivation of access to work	−0.02	0.982
Deprivation of contact with other inmates	−2.25	**0.025**
Deprivation of contact with prison officers	−3.29	**0.001**
The limited access to self-development services	−0.20	0.841
Deprivation of liberty	−0.80	0.424
The limited access to voluntary service	0.19	0.852
Physical health deterioration	−3.15	**0.002**
Mental health deterioration	−2.72	**0.007**

Explanations: *p*—level of statistical significance, *t*-test—test for two independent samples, inmates1—inmates who got sick with COVID-19, inmates2—inmates who did not experience the sickness. Bold font means that result is significant at the level of statistical significance.

**Table 4 ijerph-20-04856-t004:** Descriptive statistics for the General Mood Scale in the group of inmates (N = 250).

Items Describing the General Mood	M	SD	Skewness	Kurtosis
T. 1. I am in a bad mood	2.57	1.18	0.02	−0.35
T. 2. I feel great	2.78	1.24	−0.34	−0.17
T. 3. I am in a bad mood	2.56	1.20	−0.02	−0.28
T. 4. I feel relaxed and calm	2.86	1.23	−0.57	−0.05
T. 5. I feel gray and hopeless	2.53	1.23	−0.12	−0.49
T. 6. I have a good humour	3.01	1.19	−0.54	0.20
T. 7. I am cheerful	3.05	1.22	−0.75	0.49
T. 8. I feel depressed	2.61	1.24	−0.02	−0.32
T. 9. My well-being is bad	2.58	1.23	−0.01	−0.30
T. 10. My well-being is good	3.06	1.19	−0.55	0.20
Positive mood	2.74	2.54	0.72	−0.45
Negative mood	2.51	2.93	0.93	−0.27
General mood	27.38	6.60	−0.95	−0.76

**Table 5 ijerph-20-04856-t005:** Differences between mean values for the General Mood Scale obtained in the group of inmates who were (N = 109) and were not infected with SARS-CoV-2 (N = 141).

Items Describing the General Mood	*t*-Test	*p*
Positive mood	−1.53	0.127
Negative mood	1.96	**0.046**
General mood	−1.24	0.217

Bold font means that result is significant at the level of statistical significance.

**Table 6 ijerph-20-04856-t006:** Descriptive statistics for the Emotions Questionnaire (N = 250) for inmates.

Feelings	M	SD	Skewness	Kurtosis
joy *	3.50	1.67	−0.55	−0.20
angst	3.29	1.59	−0.19	−0.36
fear	2.83	1.70	0.21	−0.62
love *	3.56	2.07	−0.01	−0.90
anger	2.87	1.60	0.06	−0.46
cheerfulness *	3.72	1.64	−0.41	0.04
humiliation	2.45	1.66	0.76	−0.15
sadness	3.46	1.64	−0.15	−0.40
agitation	3.01	1.60	0.07	−0.20
anxiety	2.87	1.70	0.36	−0.53
grief	3.10	1.69	0.21	−0.38
concern	3.22	1.60	−0.02	−0.28
devotion *	3.07	1.82	0.13	−0.54
anger	3.04	1.58	−0.12	−0.53
breakdown	2.73	1.71	0.23	−0.66
happiness *	3.28	1.77	−0.03	−0.34
rage	2.76	1.46	0.01	−0.44
contentment *	3.65	1.67	−0.07	0.02
tenderness *	3.36	1.85	0.13	−0.55
shame	2.72	1.62	0.49	−0.03
guilt	3.29	1.80	0.01	−0.78
attachment *	3.49	2.02	0.12	−0.85
depression	2.53	2.02	0.83	−0.55
unhappiness	2.78	1.91	0.53	−0.73
positive feeling	27.71	10.51	−0.41	0.86
negative feelings	46.88	18.87	−0.25	−0.17

* positive emotions.

**Table 7 ijerph-20-04856-t007:** Descriptive statistics for the main emotions in the group of inmates (N = 250).

Main Feelings	M	SD	Skewness	Kurtosis
HAPPINESS	14.15	5.79	−0.16	0.54
LOVE	13.50	6.33	0.01	−0.05
FEAR	12.22	5.68	−0.05	−0.27
ANGER	11.67	5.37	−0.27	−0.64
GUILT	11.57	5.50	0.93	−0.22
SADNESS	11.50	6.36	0.37	−0.55

In this and the following tables and descriptions, the main emotions are written in capital letters.

**Table 8 ijerph-20-04856-t008:** Differences between the values of the mean for the main emotions in the group of inmates who were (N = 109) and were not infected with SARS-CoV-2 (N = 141).

Main Feelings	*t*-Test	*p*
HAPPINESS	−0.75	0.454
LOVE	−0.71	0.479
FEAR	−0.23	0.817
ANGER	−0.75	0.454
GUILT	−0.20	0.844
SADNESS	−0.56	0.575

**Table 9 ijerph-20-04856-t009:** Linear regression results, the dependent variable of negative mood in inmates.

INDEPENDENT VARIABLE INDICATOR Inmates2	Negative Mood *R* = 0.735; *R^2^* = 0.540; *cR^2^* = 0.439; *F* (25,141) = 5.348; *p* < 0.001			
	*Β*	*β*	*t*	*p*
Constant	−0.343		−0.327	0.744
age	0.026	0.960	1.309	0.193
joy *	−0.617	−0.311	−2.657	**0.009**
angst	0.395	0.200	1.850	0.067
fear	0.109	0.058	0.515	0.608
love *	−0.009	0.006	0.065	0.949
anger	−0.170	−0.086	−0.773	0.441
cheerfulness *	−0.010	−0.006	−0.046	0.964
humiliation	−0.056	−0.029	−0.258	0.797
sadness	0.151	0.077	0.684	0.495
agitation	0.333	0.172	1.391	0.167
anxiety	−0.198	−0.102	−0.783	0.435
grief	0.190	0.103	0.807	0.421
concern	0.282	0.141	1.101	0.273
devotion *	−0.057	−0.031	−0.327	0.745
anger	−0.219	−0.112	−0.915	0.362
breakdown	0.203	0.111	0.954	0.342
happiness *	0.121	0.069	0.691	0.491
rage	−0.013	−0.006	−0.051	0.960
contentment *	−0.175	−0.094	−0.923	0.358
tenderness *	−0.042	0.025	0.241	0.810
shame	0.214	0.069	1,069	0.287
guilt	−0.116	−0.068	−0.678	0.499
attachment *	−0.114	0.071	0.675	0.501
depression	0.247	0.154	1.318	0.190
unhappiness	0.054	0.033	0.256	0.798
**INDEPENDENT VARIABLE INDICATOR** **Inmates1**	**Negative mood** ***R* = 0.801; *R^2^* = 0.641; *cR^2^* = 0.518; *F* (25,109) = 5.222; *p* < 0.001**			
	** *Β* **	** *β* **	** *t* **	** *p* **
Constant	−3.289		−1.753	0.084
age	0.048	0.247	3.120	**0.003**
joy *	0.137	0.090	0.725	0.470
angst	−0.163	−0.097	−0.827	0.411
fear	−0.215	−0.136	−1.040	0.301
love *	−0.100	−0.078	−0.741	0.461
anger	0.249	0.149	1.512	0.134
cheerfulness *	−0.586	−0.358	−2,676	**0.009**
humiliation	−0.133	−0.085	−0.642	0.523
sadness	0.449	0.271	1.874	0.065
agitation	0.047	0.027	0.211	0.833
anxiety	−0.011	−0.007	−0.050	0.960
grief	−0.104	−0.066	−0.512	0.610
concern	0.611	0.393	2.722	**0.008**
devotion *	0.071	0.048	0.462	0.645
anger	0.309	0.181	1.385	0.170
breakdown	−0.078	−0.051	−0.392	0.696
happiness *	−0.059	−0.039	−0.335	0.738
rage	−0.493	−0.266	−2.064	**0.042**
contentment *	0.227	0.145	1.084	0.281
tenderness *	0.180	0.121	0.945	0.347
shame	−0.233	−0.141	−1.368	0.175
guilt	−0.309	−0.202	−1.683	0.096
attachment *	−0.144	−0.108	−1.683	0.332
depression	0.121	0.090	−0.998	0.534
unhappiness	0.695	0.501	0.625	**0.001**

Explanations: Inmates1—inmates who were infected with SARS-CoV-2; Inmates2—inmates who were not infected with SARS-CoV-2. Bold font means that result is significant at the level of statistical significance. * positive emotions.

**Table 10 ijerph-20-04856-t010:** Linear regression results, the dependent variable of positive mood in inmates.

INDEPENDENT VARIABLE INDICATOR Inmates2	Positive Mood *R* = 0.800; *R^2^* = 0.640; *cR^2^* = 0.561; *F* (25,141) = 8,091; *p* < 0.001			
	*Β*	*β*	*t*	*p*
Constant	3.242		4.231	0.001
age	0.004	0.016	0.241	0.810
joy *	0.356	0.218	2,102	**0.038**
angst	−0.378	−0.232	−2.427	**0.017**
fear	−0.063	−0.041	−0.409	0.683
love *	−0.024	−0.204	−0.232	0.817
anger	0.192	0.117	1.199	0.233
cheerfulness *	0.186	0.119	1.114	0.268
humiliation	−0.059	−0.037	−0.376	0.708
sadness	−0.309	−0.190	−1.915	0.058
agitation	−0.164	−0.103	−0.938	0.350
anxiety	0.105	0.066	0.572	0.538
grief	0.133	−0.087	−0.772	0.442
concern	−0.316	−0.191	−1.688	0.094
devotion *	0.402	0.028	0.327	0.744
anger	−0.145	−0.089	−0.828	0.409
breakdown	0.079	0.052	0.510	0.611
happiness *	0.099	0.068	0.773	0.441
rage	−0.240	−0.146	−1.330	0.186
contentment *	0.335	0.218	2.425	**0.017**
tenderness *	−0.165	−0.118	−1.305	0.194
shame	0.053	0.033	0.361	0.719
guilt	0.048	0.034	0.386	0.700
attachment *	0.069	0.052	0.562	0.575
depression	0.062	0.047	0.450	0.653
unhappiness	0.007	0.005	0.047	0.962
**INDEPENDENT VARIABLE INDICATOR** **Inmates1**	**Positive mood** ** *R* ** **= 0.788; *R^2^* = 0.621; *cR^2^* = 0.492; *F* (25,109) = 4,796; *p* < 0.001**			
	** *Β* **	** *β* **	** *t* **	** *p* **
Constant	6.369		3.472	0.001
age	0.003	0.015	0.189	0.851
joy *	0.022	0.015	0.122	0.904
angst	−0.134	−0.084	−0.698	0.487
fear	−0.144	−0.096	−0.714	0.477
love *	0.028	0.023	0.212	0.833
anger	0.001	0.000	0.001	0.999
cheerfulness *	0.284	0.183	1,328	0.188
humiliation	0.210	0.124	1.043	0.300
sadness	−0.221	−0.140	−0.942	0.349
agitation	0.041	0.024	0.187	0.852
anxiety	−0.044	−0.031	−0.217	0.828
grief	−0.233	−0.154	−1.168	0.246
concern	0.326	0.220	1.485	0.141
devotion *	−0.229	−0.164	−1.525	0.131
anger	0.153	0.094	0.702	0.485
breakdown	−0.376	−0.258	−1.932	0.057
happiness *	0.423	0.295	2.455	**0.016**
rage	−0.362	−0.205	−1.552	0.125
contentment *	−0.020	−0.014	−0.100	0.921
tenderness *	0.238	0.069	1.282	0.204
shame	−0.003	−0.002	−0.016	0.988
guilt	−0.027	−0.018	−0.149	0.882
attachment *	−0.195	−0.153	−1.383	0.171
depression	−0.084	−0.066	−0.441	0.660
unhappiness	−0.120	−0.091	−0.638	0.525

* positive emotions.

## Data Availability

Data supporting the reported results can be obtained by email from the corresponding author.

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
