# Peer review of "Mood and Emotions among Inmates after COVID-19 Pandemic"

_ijerph, 2023, doi:10.3390/ijerph20064856_

Round 1

Reviewer 1 Report

The purpose of the reviewed study was to assess moods and emotions experienced by prisoners and Prison Service officers after the period of the pandemic and to compare both groups.

It is worthy to notice that this research was conducted in representative group prison officers (N=250) and prisoners (N=250) in six randomly selected Polish prisons.

Apart from that the research problem raises some objections:

1.      In my opinion this research assesses moods and emotions which are currently experienced and putting them in the context of the pandemic period is too broad, especially that the scales measure emotions and mood which are “experienced by an individual at the time of the survey (252), and over the past week” (253).

2.      There was no additional survey dedicated to the pandemic period and the questions from three standardized survey instruments used in research related to just “general mood” (263) and “emotions” (319). We have no direct pandemic context in this research.

3.      As Authors say, we can’t compare possible changes in the perceived emotions and mood (509-510) because of the lack of the longitudinal study.

In my opinion this research assesses mood and emotions currently experienced by prisoners (1 group) and prison service officers (2 group) in the context of imprisonment (1 group) and hard, burning-work in prison (2 group). There are enough factors which are sufficient to induce negative emotions and mood in both groups (which are mentioned by Authors in Discussion section) and Authors could guide theirs considerations in this direction. Therefore in Introduction section Authors could pay more attention to stressful circumstances of imprisonment than to pandemic period.

So, I think that this article is just about: Mood and emotions among Prison Service officers and inmates after the pandemic. To much generalization of results is a mistake.

Next issue relates to Materials – could you, please, give more information about overall score in scales? “The scales are characterized by satisfactory psychometric properties” (251) – namely? What is reliability and validity scores of this scales?

Next issue relates to Results:

1.      To be honest, I do not understand why Authors compare those two groups: prisoners (1 group) and prison service officers ( 2 group)? Isn’t is obvious that officers would have better mood and much mode positive emotions than prisoners because the latter are incarcerated? Why is it so important to compare those groups? Please, explain more this issue.

2.      Why Authors compare items rather than overall score in the General Mood Scale? (Table 4)

3.      Is there anything surprising in the results of conducted research? Something non-obvious?

4.      The most interesting result is in regression analysis – It would be worthy if Authors could discuss those models deeper:

·        Why depression, happiness and sadness predict mood in prison service and why anger and love  predict mood of inmates? 

·        why these variables explain low percentage (9% and 10%) of the variance? Which other factors could predict mood in those two groups and why?

Concluding, In my opinion this article is noteworthy, but it needs corrections, starting with problem research and in consequences – with conclusions and discussion.

Thank you for your attention!

Author Response

Thank you so much for your effort and valuable suggestions to improve the paper.

Reviewer 2 Report

I like the general idea of this study, as it addresses the important topic. Another strong point is its sample (inmates are not an easy population to reach, and the authors did a really good job at obtaining this sample).

However, I am not sure the study really addresses the issue it claims to address: the mood in relation to the pandemic. While there is no doubt that the pandemic has affected all areas of our lives, we do not have any data regarding the mood of inmates (or officers) prior to the pandemic. As the authors stated, the study was conducted in 2022, when restrictions were no longer in place, and the inmates were allowed to have visitors.

I was hoping that the authors are going to address this question at least indirectly, through the COVID-related data they have collected, potentially as covariates when comparing the two groups, but they have only compared the scores of inmates and officers, without taking account the COVID experiences. (If, by any chance, the authors are in possession of data on the same scales prior to the pandemic – I would definitively encourage them to present that comparison as well.) As it is, these results might have been the same where they obtained in 2018 (it would still be logical to expect better mood among officers than inmates) – and I would suggest that the authors adjust their discussion and especially conclusions accordingly.

Furthermore, the reported analyses do not reflect the aim and the hypotheses described at the end of the introduction. So, suddenly, in table 10 (line 377) we are presented with a regression analyses – which does not stem from any of the announced research questions and or hypotheses. (more comments regarding this particular analysis later on).

Regarding the sample: 20% of the officers are women, while there were no women in the inmate group. Gender is an important variable here, and should be controlled for in the analyses. At the very least, try to re-run all the analyses without the – women in the officers’ sample, and see if this changes the results.

Regarding the procedure: The authors state: “The survey was preceded by instructions from the researchers on how to conduct the survey.” – Please provide more information: was it a group testing, or did the prison psychologist individually approach and test each inmate?

The instruments:

Please provide references for the instruments. (for example, line 250 – B. Wojciszke and W. Barilla… is there a year for this publication, or any kind of reference for these instruments? If no, please cite the “unpublished thesis…” or “personal communication” in whatever is the appropriate style for this journal)

Line: 251: “.. characterized by satisfactory psychometric 251 properties.” – We believe you, but please provide them (at least the factorial structure and alphas for the scales)

The above comment refers to all the instruments, as the psychometric properties for any of them were not reported.  

The results:

What was the rationale for using separate items for analysis? From the mood scale items, it seems it comprises from at least two sub-scales (positive and negative moods) – why not compute an average score?

Table 4, column “group” is redundant: these are the only two groups you are comparing, and you are not reporting Ms and SDs, which would be different for the two groups, so no need to state that 10 times.

Line 295/6: there is some repetition here: “They were in a depressed mood 295 (M1=1.94, M2=2.57), bad mood (M1=1.95, M2=2.56), and bad mood (M1=1.97, M2=2.58).”

The stepwise regression: please, report the B,p  and other values for ALL variables used in the model, not just the ones with significant beta ponders. If the analysis was indeed stepwise, please report the blocks entered (sets of variables entered in the same step), and changes in the proportion of variance explained by each step.  Also, where is the variable MOOD in this particular analysis derived from? Is it a composite measure of the General mood scale? Please specify.

This is, in my opinion the most informative analysis in this study, and it was given far less space than the numerous, basically descriptive analyses preceding it.  

Minor comments:

Line 233: “…Juveniles up to the age of 24 accounted for 11% “ - is this the correct term? Juveniles is the term usually reserved for minors, so it rings a bit unusual.

“in the middle of 2022” – this is an odd phrase. If it was conducted in June/July, etc., please state so

Line 282: “The result can be interpreted as moderate mood. “  - Unfortunately, this statement doesn’t mean much. One can be e.g. moderately cheerful, or moderately depressed, but what does it mean if one is in a moderate mood?

Author Response

Please see the attachment. Thank you so much for your effort and valuable suggestions to improve the paper.

Round 2

Reviewer 1 Report

I would like to express my admiration for Authors’ work aiming at improvement the article. Now, it is much better and – in my opinion – much more interesting, when you compare 2 groups of prisoners.   Thank you for inclusion my suggestion and request. I have no more comments, everything is correct.

I recommend the article for publication.

Author Response

Response to Reviewer 1 Comments

Point 1: I would like to express my admiration for Authors’ work aiming at improvement the article. Now, it is much better and – in my opinion – much more interesting, when you compare 2 groups of prisoners. Thank you for inclusion my suggestion and request. I have no more comments, everything is correct. 

Response 1: We are gratefull for your suggestions and comments. The manuscript in the present form includes all the necessary changes pointed out by Reviewers to emprove its quality. Thank you so much.

Reviewer 2 Report

In the new abstract, authors state that they performed „feature selection analysis “. This is a very specific term and it is unclear what do the authors refer to? Because regression analysis is not feature selection analysis.

When describing the contribution of specific predictors to the overall proportion of explained variance, please DO NOT use p values as indicators of their importance. Beta values are the ones that indicate the size of the contribution, p-values just estimate the probability that these findings are accidental, they DO NOT say anything about the size of the effect. This comment refers to both the abstract and the results section – and any other part of the text where p is used as a descriptor of variable’s “importance”. (For example: “Regression analysis showed that the variable that had the greatest impact on inmates' positive mood were felt happiness (p<0.05) (for 28 those who during imprisonment got sick with Covid-19) and joy (p<0.05), angst (p<0.05) and content 29 (p<0.05) (for healthy ones)”)

Please, refrain from using phrases such as: “Therefore, the Student’s t-test for independent samples was applied to confirm this scientifically”. It is not specific data analysis that makes a research study more or less “scientific” – it is the methodology, the theoretical rationale, the reasoning behind the interpretation of results: one cannot just apply t-test to provide a provide a “scientific” confirmation.

Again, the descriptions of the tables are not accurate: for example, the authors state that “Table 8 shows the values of: mean, standard deviation, Student’s t-test, and statistical significance for the main emotions.” When in fact, what we see in the first row of this table are: F-values, p-values, t-test values, and again p-values.

(it is also unclear why shoe BOTH the F-values and t values.)

As for tables 9 and 10:

-       - there are two columns labelled as “B” – one of those columns probably contains standardized and the other one unstandardized beta coefficients. These two carry different labels: please use the Greek letter β for the standardized coefficients.

-    - what do the asterisk denote? They are put randomly on some significant and some insignificant predictors?

Author Response

Response to Reviewer 2 Comments

Point 1: In the new abstract, authors state that they performed „feature selection analysis “. This is a very specific term and it is unclear what do the authors refer to? Because regression analysis is not feature selection analysis.

Response 1: The sentence “Correlational and feature selection analysis was conducted on quantitative data” is changed for “Comparative and regression analyses were carried out”.

Point 2: When describing the contribution of specific predictors to the overall proportion of explained variance, please DO NOT use p values as indicators of their importance. Beta values are the ones that indicate the size of the contribution, p-values just estimate the probability that these findings are accidental, they DO NOT say anything about the size of the effect. This comment refers to both the abstract and the results section – and any other part of the text where p is used as a descriptor of variable’s “importance”. (For example: “Regression analysis showed that the variable that had the greatest impact on inmates' positive mood were felt happiness (p<0.05) (for 28 those who during imprisonment got sick with Covid-19) and joy (p<0.05), angst (p<0.05) and content 29 (p<0.05) (for healthy ones)”)

Response 2: All indicated mistakes have been corrected in the abstract, the results section and descriptions below the regression results tables.

Point 3: Please, refrain from using phrases such as: “Therefore, the Student’s t-test for independent samples was applied to confirm this scientifically”. It is not specific data analysis that makes a research study more or less “scientific” – it is the methodology, the theoretical rationale, the reasoning behind the interpretation of results: one cannot just apply t-test to provide a provide a “scientific” confirmation.

Response 3: All descriptions have been corrected as suggested avoiding their 'scientific' confirmation.

Point 4: Again, the descriptions of the tables are not accurate: for example, the authors state that “Table 8 shows the values of: mean, standard deviation, Student’s t-test, and statistical significance for the main emotions.” When in fact, what we see in the first row of this table are: F-values, p-values, t-test values, and again p-values. (it is also unclear why shoe BOTH the F-values and t values.)

Response 4: Descriptions of the Tables 3,5,8 are corrected - only descriptions to the data in the tables are left. F-test values and significance levels in these tables have been removed. Only t-test values and Significance Levels for this test have been left.

Point 5: As for tables 9 and 10:

 -  there are two columns labelled as “B” – one of those columns probably contains standardized and the other one unstandardized beta coefficients. These two carry different labels: please use the Greek letter β for the standardized coefficients.

-what do the asterisk denote? They are put randomly on some significant and some insignificant predictors?

Response 5: - Yes, you are right one of the values is β. Now it is corrected.

  • Explanations: * positive emotions. An asterisk explanation is given below Table 6. It does not indicate statistically significant results.
